# Polygenic risk scores for the prediction of common cancers in East Asians: A population-based prospective cohort study

Peh Joo Ho[1,2,3], Iain BeeHuat Tan[1,2,4,5], Dawn Qingqing Chong[5,6], Chiea Chuen Khor[1], Jian-Min Yuan[7,8], Woon-Puay Koh[9,10], Rajkumar Dorajoo[1]*, Jingmei Li[1,3]*

[1]Genome Institute of Singapore (GIS), Agency for Science, Technology and Research (A*STAR), Singapore, Singapore; [2]Saw Swee Hock School of Public Health, National University of Singapore and National University Health System, Singapore, Singapore; [3]Department of Surgery, Yong Loo Lin School of Medicine, National University of Singapore, Singapore, Singapore; [4]Program in Cancer and Stem Cell Biology, Duke-National University of Singapore Medical School, Singapore, Singapore; [5]Division of Medical Oncology, National Cancer Centre Singapore, Singapore, Singapore; [6]Duke-NUS Medical School Singapore, Singapore, Singapore; [7]UPMC Hillman Cancer Center, Pittsburgh, United States; [8]Department of Epidemiology, University of Pittsburgh Graduate School of Public Health, Pittsburgh, United States; [9]Healthy Longevity Translational Research Programme; Yong Loo Lin School of Medicine, National University of Singapore, Singapore, Singapore; [10]Singapore Institute for Clinical Sciences, Agency for Science Technology and Research (A*STAR), Singapore, Singapore

*For correspondence:
dorajoor@gis.a-star.edu.sg (RD);
lijm1@gis.a-star.edu.sg (JL)

## Abstract

**Background:** To evaluate the utility of polygenic risk scores (PRSs) in identifying high-risk individuals, different publicly available PRSs for breast (n=85), prostate (n=37), colorectal (n=22), and lung cancers (n=11) were examined in a prospective study of 21,694 Chinese adults.

**Methods:** We constructed PRS using weights curated in the online PGS Catalog. PRS performance was evaluated by distribution, discrimination, predictive ability, and calibration. Hazard ratios (HR) and corresponding confidence intervals (CI) of the common cancers after 20 years of follow-up were estimated using Cox proportional hazard models for different levels of PRS.

**Results:** A total of 495 breast, 308 prostate, 332 female-colorectal, 409 male-colorectal, 181 female-lung, and 381 male-lung incident cancers were identified. The area under receiver operating characteristic curve for the best-performing site-specific PRS were 0.61 (PGS000873, breast), 0.70 (PGS00662, prostate), 0.65 (PGS000055, female-colorectal), 0.60 (PGS000734, male-colorectal), 0.56 (PGS000721, female-lung), and 0.58 (PGS000070, male-lung), respectively. Compared to the middle quintile, individuals in the highest cancer-specific PRS quintile were 64% more likely to develop cancers of the breast, prostate, and colorectal. For lung cancer, the lowest cancer-specific PRS quintile was associated with 28–34% decreased risk compared to the middle quintile. In contrast, the HR observed for quintiles 4 (female-lung: 0.95 [0.61–1.47]; male-lung: 1.14 [0.82–1.57]) and 5 (female-lung: 0.95 [0.61–1.47]) were not significantly different from that for the middle quintile.

**Conclusions:** Site-specific PRSs can stratify the risk of developing breast, prostate, and colorectal cancers in this East Asian population. Appropriate correction factors may be required to improve calibration.

**Funding:** This work is supported by the National Research Foundation Singapore (NRF-NRFF2017-02), PRECISION Health Research, Singapore (PRECISE) and the Agency for Science, Technology and Research (A*STAR). WP Koh was supported by National Medical Research Council, Singapore (NMRC/CSA/0055/2013). CC Khor was supported by National Research Foundation Singapore (NRF-NRFI2018-01). Rajkumar Dorajoo received a grant from the Agency for Science, Technology and Research Career Development Award (A*STAR CDA - 202D8090), and from Ministry of Health Healthy Longevity Catalyst Award (HLCA20Jan-0022).

The Singapore Chinese Health Study was supported by grants from the National Medical Research Council, Singapore (NMRC/CIRG/1456/2016) and the U.S. National Institutes of Health (NIH) (R01 CA144034 and UM1 CA182876).

## Editor's evaluation

Utilizing one of the largest prospective Asian cohorts with long-term follow-up data, the important study reveals the utility of polygenic risk scores in identifying high-risk individuals from specific cancers. The convincing results reached by the high-quality data highlight the translational significance of this study, which may be important for future exploration in the field of cancer epidemiology.

## Introduction

Polygenic risk scores (PRSs) for a range of health traits and conditions have been developed in recent years. These scores, which are based on summary statistics from genome-wide association studies (GWAS), can be used to stratify people depending on their genetic risk of acquiring various diseases, to improve screening and preventative interventions, as well as patient care (*Polygenic Risk Score Task Force of the International Common Disease Alliance, 2021*; *Lambert et al., 2019*). Precision risk assessment may help develop tailored screening strategies targeting individuals at higher risk of disease of interest (*Clift et al., 2022*).

The contributions of heritable genetic factors are different for different cancers. Twin studies have highlighted statistically significant effects of heritable genetic risk factors for cancers of the prostate, colorectal, and breast (*Lichtenstein et al., 2000*). The amount of phenotypic variance explained by the common genetic variants found by GWAS is also known to vary (*Cano-Gamez and Trynka, 2020*), suggesting that PRS derived from GWAS findings may perform to varying degrees for different cancers.

The area under receiver operating characteristic curve (AUC) is an important discrimination index for evaluating the performance of PRS. The greater the AUC, the better the discriminatory ability to separate cases from non-cases. A value of 0.5 suggests that the tool is performing no better than chance, while a value of 1 is obtained when cases and non-cases are perfectly separated. The range of reported AUC associated with published PRS ranged from 0.584 to 0.678 for breast cancer (*Mavaddat et al., 2019*; *Ho et al., 2020*; *Kachuri et al., 2020*; *Du et al., 2021*; *Jia et al., 2020*; *Lacaze et al., 2021*; *Zhang et al., 2018*), 0.591–0.769 for prostate cancer (*Kachuri et al., 2020*; *Jia et al., 2020*; *Fritsche et al., 2020*), 0.609–0.708 for colorectal cancer (*Kachuri et al., 2020*; *Jia et al., 2020*; *Gafni et al., 2021*; *Archambault et al., 2022*), and 0.52–0.846 for lung cancer (*Kachuri et al., 2020*; *Jia et al., 2020*; *Fritsche et al., 2020*; *Hung et al., 2021*). In a study by Jia et al. looking at eight common cancers in the UK Biobank population-based cohort study (n=400,812 participants of European descent), the observed AUC ranged from 0.567 to 0.662 (*Jia et al., 2020*).

While prediction of individual cancer risks through PRS remains moderate, emerging data supports the use of PRS for population-based cancer risk stratification. In previous work, Ho et al. examined the overlap of women identified to be at high risk of developing breast cancer based on family history for the disease, a non-genetic breast cancer risk prediction model, a breast cancer PRS, and carriership of rare pathogenic variants in established breast cancer predisposition genes (*Ho et al., 2022*). The overlap of individuals found to be at elevated risk of developing breast cancer based on the genetic

**eLife digest** Although humans contain the same genes, the sequence within these DNA sites can vary from person to person. These small variations, also known as genetic variants, can increase the risk of developing certain diseases. While each variant will only have a weak effect, if multiple variations are present the odds of developing the disease becomes significantly higher.

To determine which variants are linked to a disease, researchers carry out genome-wide association studies which involve analyzing the genomes of individuals with and without the condition and comparing their genetic codes. This data is then used to calculate how different combinations of variants impact a person's chance of getting the disease, also known as a polygenic risk score.

Currently, most genome-wide association studies only incorporate genetic data from people with European ancestry. Consequently, polygenic risk scores performed using this information may not accurately predict the risk of developing the disease for individuals with other ethnicities, such as people with Asian ancestry.

Here, Ho et al. evaluated how well previously calculated polygenic risk scores for the four most common cancers (breast, colorectal, prostate and lung) worked on individuals of East Asian descent. The scores were tested on a dataset containing the genetic sequence, medical history, diet and activity levels of over 21,000 people living in Singapore in the 1990s. Ho et al. found that the polygenic risk scores for breast, prostate and colorectal cancer were able to predict disease risk. However, the score for lung cancer did not perform as well. The polygenic risk score for breast cancer was the most accurate, and was able to stratify individuals into distinct risk bands at an earlier age than other scores.

These findings shed light on which existing polygenic risk scores will be effective at assessing cancer risk in individuals with East Asian ancestry. Indeed, Ho et al. have already incorporated the polygenic risk score for breast cancer into a pilot study screening individuals in a comparable population in Singapore. However, the polygenic risk scores tested still performed better on individuals with European ancestry, highlighting the need to address the lack of Asian representation in genome-wide association studies.

and non-genetic models was low. PRS was also found to be able to identify high-risk individuals among young women who were not yet eligible to attend mammography screening. The findings suggest that a genetic tool that is feasible to be deployed for population-based screening may complement current screening programs.

Disparities in the genetic risk of cancer among various ancestry populations are poorly understood. Ideally, selected genetic variants that make up PRS should be relevant to the population being screened. The development of training datasets of PRS are dominated by samples of European ancestry, resulting in ancestry bias and issues with transferability to other populations (*Lambert et al., 2019*; *Fritsche et al., 2021*). The mismatch between the ancestries of the GWAS samples and the target populations for PRS application is a limiting factor (*Fritsche et al., 2021*). In this study, we evaluated the utility of common PRS, curated in the Polygenic Score (PGS) Catalog, in predicting the risk of the commonly diagnosed cancers with high genetic predisposition (breast, prostate, colorectal, and lung) in a prospective cohort comprising 21,694 participants of East Asian descent in Singapore. The reporting framework recommended for the interpretation and evaluation of PRS detailed in Wand et al. is used (*Wand et al., 2021*).

## Materials and methods

### Singapore Chinese Health Study

The Singapore Chinese Health Study (SCHS) is a population-based prospective cohort study of ethnic Chinese men and women recruited between April 1993 and December 1998 (*Hankin et al., 2001*). Participants were 45–74 years of age at recruitment and were restricted to the two major dialect groups of Chinese adults in Singapore, who were the Hokkiens and the Cantonese that had originated from Fujian and Guangdong provinces in Southern China, respectively. All our study participants were residents of government housing flats, which were built to accommodate approximately 86% of the resident population in Singapore during the enrolment period. A total of 63,257 individuals

(35,298 women and 27,959 men) provided written informed consent (**Hankin et al., 2001**). The study was approved by the Institutional Review Boards of the National University of Singapore, University of Pittsburgh, and the Agency for Science, Technology and Research (A*STAR, reference number 2022-042). Written, informed consent was obtained from all study participants.

## Baseline

An in-person baseline interview was performed at recruitment to collect data on diet using a validated 165-item food frequency questionnaire, smoking, alcohol, physical activity, medical history, and menstrual and reproductive history from women.

## Selection of common cancers

In Singapore, between 2015 and 2019, colorectal cancer, the most prevalent cancer in men, accounted for nearly 17% of cancer diagnoses, while breast cancer, the most common cancer in women, accounted for about three out of ten cancer diagnoses (**National Registry of Diseases Office, 2021**). During this time, cancers of the breast, prostate, colorectal, and lung accounted for approximately half of the total cancer diagnoses. These four most common cancers were selected for inclusion in this study. We further stratified the analysis by sex as differences in colorectal and lung cancer incidence by sex have been reported in Singapore (**de Kok et al., 2008**).

A unique National Registration Identity Card (NRIC) number for every Singaporean enables the compilation and linkage of data from national register data to the same individual (**Emmanuel, 1993**). Identification of incident cases of cancer was accomplished by record linkage of all surviving cohort participants with the database of the nationwide Singapore Cancer Registry (**Emmanuel, 1993**). The Singapore Cancer Register was founded in 1968. Prior to 2009, reporting of neoplasms by all medical practitioners and pathology laboratories to the registry is voluntary (**Fung et al., 2016**). The registry's staff compares cancer patient hospital discharges and death certificates to registered cases for verification. Completeness of reporting in the 1970s is 96% and in the 1990s, it was close to 100% (**Sim et al., 2006**). Cancers that developed among SCHS participants were identified using International Classification of Diseases (ICD) codes ICD-O-3 (breast: C50, prostate: C61, colorectal: C18, C19, and C20, lung: C34).

## Follow-up

Death date was obtained by record linkage with the database Birth and Death Registry of Singapore (**Emmanuel, 1993**). The data on migration in our cohort was collected during our subsequent follow-up interviews, and we were informed about the migration by the family members of cohort members who had migrated. To date, only 47 (<1%) of the entire cohort participants were known to be lost to follow up due to migration out of Singapore, suggesting that the ascertainment of cancer and death incidences among the cohort participants was virtually complete.

## Genotyping and imputation

Between 1999 and 2004, a total of 28,346 subjects contributed blood samples. A total of 25,273 SCHS participants were genotyped between the years 2017–2018 with the Illumina Infinium Global Screening Array (GSA) v1.0 and v2.0 (**Chang et al., 2021**).

Details on the sample quality control (QC) processes are previously described (**Chang et al., 2021**). Briefly, samples with a call rate of 95% or below (n=176) or heterozygosity extremes (>3 standard deviation [SD], n=236) were removed. Identity-by-state measurements were performed by pairwise comparisons of samples to detect related samples (first and second degree). One sample from each identified pair with the lower call rate was eliminated from further analysis (n=2746). To identify any ethnic outliers, principal component analysis was used in conjunction with 1000 Genomes Project reference populations and within the SCHS samples, which resulted in the further removal of 287 samples. Of the 21,828 samples that passed genotyping QC, 134 participants who were diagnosed with cancer before recruitment or had missing cancer outcomes and were excluded from the study, resulting in a final analytical dataset of 21,694 (**Supplementary file 1a**).

Alleles for all SNPs were coded to the forward strand and mapped to hg19. SNP QC steps included the exclusion of sex-linked and mitochondrial variants, gross Hardy–Weinberg equilibrium (HWE) outliers ($p<1 \times 10^{-6}$), monomorphic SNPs, or those with a minor allele frequency (MAF) < 1.0%,

and SNPs with low call rates (<95.0%). We imputed for additional autosomal SNPs using IMPUTE v2 (*Marchini et al., 2007*) and with a two reference panel imputation approach by including (1) the cosmopolitan 1000 Genomes reference panels (Phase 3, representing 2504 samples) and (2) an Asian panel comprising 4810 Singaporeans (2780 Chinese, 903 Malays, 1127 Indians) (*Chang et al., 2021*). SNPs with imputation quality score INFO <0.8, MAF <1.0%, or HWE p<1 × 10⁻⁶, as well as non-biallelic SNPs were excluded.

## Polygenic risk scores

Published PRSs were retrieved from the PGS Catalog, an open database of polygenic scores (retrieved on 26 February 2022) (*Supplementary file 1b*; *Lambert et al., 2021*). Of the 2166 PRSs available in the resource, 1706 PRSs comprising less than 100,000 predictors were downloaded. Only PRSs with odds ratios or log odds ratios as weights were included; we excluded PRSs which used odds ratio over expected risk, inverse-variance weighting, and unweighted. A total of 85, 37, 22, and 11 PRSs were available for breast, prostate, colorectal, and lung cancers, respectively. *Supplementary file 1c* shows the number of individual variants comprising each PRS and proportion of variants missing in the SCHS cohort. Individual PRSs were calculated using the allelic scoring (–score sum) functions with default parameters in PLINK (v1.90b5.2) (*Chang et al., 2015*). The formula used was,

$$PRS = \beta_1 x_1 + \beta_2 x_2 + \ldots + \beta_k x_k + \ldots + \beta_{313} x_{313}$$

where $x_k$ is the dosage of risk allele (0–2) for SNP $k$, $\beta_k$ is the corresponding weight.

## PRS distribution

Two-sided, two-sample t-tests with a type I error of 0.05 were used to examine whether there was a difference in the distribution of standardised PRS (subtraction of mean value followed by the division by the SD) between site-specific cancer cases and non-cancer controls.

## PRS discrimination

Discrimination was quantified by the area under the receiver operating characteristic curve (AUC), using logistic regression models, and their corresponding 95% CI. An AUC of 0.9–1.0 is considered excellent, 0.8–0.9 very good, 0.7–0.8 good, 0.6–0.7 sufficient, and 0.5–0.6 insufficient (*Šimundić, 2009*). The site-specific PRS with the highest AUC (logistic regression models) was selected. To test the sensitivity of the PRS selection, we obtained a time-to-event metric for AUC at 5 years, we used AUC.cd() from the 'survAUC' package in R (*Kamarudin et al., 2017*).

## Associations between PRS and risk of developing cancers

Subjects were classified into PRS percentile groups. Person-years of follow-up were calculated for each subject from the date of enrolment to the date of cancer diagnosis, death, or 31 December 2015 (the date of linkage with the Singapore Cancer Registry), whichever came first. Follow-up time was censored at 20 years after recruitment. The associations between cancer-specific PRS quintiles (where individuals ranked by PRS were categorised into quintiles, using the middle quintile [40–60%] as a reference to reflect the average risk of the population) and the incidence of site-specific cancers were investigated using Cox proportional hazards modelling to estimate hazard ratios (HR) and corresponding 95% confidence intervals (CI), using time since recruitment as the time scale, and adjusted for age at recruitment. Tests for trends were conducted using two-sided Wald tests with a type I error of 0.05. Assumptions for proportional hazards were checked using the cox.zph() function in the 'survival' package in R, where a formal score test is done to test if a time-dependent variable is required.

HR and corresponding 95% CI were also estimated for every SD increase in PRS. PRS is known to have 'portability' issues related to genetic ancestry and demographics (*Martin et al., 2019*; *Mostafavi et al., 2020*). Hence, we adjusted for variables in the models, including age at recruitment, dialect group (Hokkien or Cantonese), highest level of education (no formal education, primary school, or secondary or higher), body mass index (continuous, kg/m²), cigarette smoking (non-smoker, ex-smoker, current smoker), alcohol consumption (never, weekly, daily), moderate physical activity (none, 1–3 hr/week, ≥3 hr/week), vigorous work/strenuous physical activity at least once a week (no or yes), and familial history of cancer (no or yes).

## PRS absolute risk association

The 5-year absolute risks of developing breast, prostate, colorectal, and lung cancers were computed for PRS groups of increasing five percentiles over the follow-up period. Incidence (between 2013 and 2017) and mortality (the year 2016) statistics in Singapore (reported in *National Registry of Disease Office, 1968* and *Singapore Statistics, 2023*, respectively) were used for the absolute risk estimations. We estimated the cancer-specific 5-year absolute risk based on PRS an iterative method detailed by *Mavaddat et al., 2015*.

## PRS calibration

Calibration was studied by comparing the expected proportion of cases in the 5 years after recruitment to the observed proportion of cases that occurred in that 5 years, within each decile of PRS. Linear regression of the 10 points (pairs of expected and observed proportion) was used to study the overall calibration. A curve close to the diagonal indicates that predicted cancer risks correspond well to observed proportions. A slope above 1 implies that the model underestimates the absolute risk. Conversely, a slope below 1 implies that the model overestimates the absolute risk. In addition, we used the Hosmer-Lemeshow test to check the goodness-of-fit.

## Results

### Characteristics of the study population

*Table 1* shows the characteristics of the 21,694 participants who were cancer-free at recruitment. The median follow-up time for the cohort was 20 years (interquartile range [IQR]: 18–22). As of December 2015, 495 women developed breast cancer, 308 men developed prostate, 774 (332 women and 409 men) colorectal cancer, and 562 (181 women and 381) lung cancer. The median age at recruitment was 54 years (IQR: 49–61). The median age at diagnosis was 65 years (IQR: 59–70) for female breast cancers, 72 years (IQR: 67–77) for prostate cancers, 71 years (IQR: 65–76) for male colorectal cancers, 71 years (IQR: 64–78) for female colorectal cancers, 74 years (IQR: 68–78) for male lung cancers, and 74 years (IQR: 66–79) for female lung cancers.

### Lack of Asian representation in PRS development

Among PRS for breast (n=85), prostate (n=37), colorectal (n=22), and lung cancers (n=11) examined, the reported source of variant associations or GWAS used to build PRS were from predominantly European ancestry populations (*Supplementary file 1c*). Only six PRS for breast cancer (PGS000028, PGS000029, PGS000050, PGS000345, PGS0001336, and PGS001804), three for prostate cancer (PGS000878, PGS001291, and PGS001805), three PRS for colorectal cancer (PGS000055, PGS000802, and PGS001802), and one for lung cancer (PGS000070) were based on GWAS that included some non-European participants. For PRS development training, all but two PRSs were based on samples of non-European ancestry (PGS000733 for prostate cancer and PGS000802 for colorectal cancer). No significant association (p>0.05) was found between number of variants included in the various PRSs evaluated for each cancer and discriminatory ability (*Supplementary file 1d*).

### PRS distribution

*Figure 1* depicts the (A) distribution, (B) discrimination, (C) absolute risk association, and (D) calibration of the best-performing PRS (based on AUC) (*Supplementary file 1d*) for the four cancers studied: breast (PGS000873; *Brentnall et al., 2020*), prostate (PGS000662; *Conti et al., 2021*), colorectal (female: PGS000055; *Schmit et al., 2019*; male: PGS000734; *Archambault et al., 2020*), and lung (female: PGS000721; *Jia et al., 2020*; male: PGS000070; *Dai et al., 2019*). All PRSs were normally distributed, with a right shift observed in the distribution curves for cancer cases (*Figure 1A*). The mean value of each site-specific cancer PRS was significantly higher in cancer patients compared to controls ($p_{t\text{-test}}$ <0.00273).

### PRS discriminatory ability

The highest AUC obtained from logistic models was observed for prostate cancer (0.70, 95% CI: [0.66–0.73]), followed by female breast cancer (0.61 [0.58–0.63]), male colorectal cancer (0.60,

**Table 1.** Demographics of our study population by gender and cancer site.

Demographics variables were collected using structured questionnaire at recruitment. Family history for lung cancer was not available. Information on cancer occurrence (number of cancer and age at cancer occurrence) was obtained through linkage with the Singapore Cancer Registry in December 2015. Follow-up time was calculated from age at recruitment. IQR: Interquartile range.

| | Entire cohort | | | Individuals who developed cancer | | | | | |
|---|---|---|---|---|---|---|---|---|---|
| | | | | Breast | Prostate | Colorectal | | Lung | |
| | All | Female | Male | Female | Male | Female | Male | Female | Male |
| n | 21,694 | 12,084 | 9610 | 495 | 308 | 332 | 409 | 181 | 381 |
| Age at recruitment in years, median (IQR) | 54 (49–61) | 54 (48–60) | 55 (49–62) | 53 (48–59) | 59 (54–64) | 58 (52–64) | 59 (52–65) | 59 (55–64) | 60 (55–64) |
| Number of cancers developed | | | | | | | | | |
| 0 (did not develop cancer) | 19633 (90) | 11096 (92) | 8537 (89) | – | – | – | – | – | – |
| 1 | 2013 (9) | 968 (8) | 1045 (11) | 476 (96) | 293 (95) | 317 (95) | 387 (95) | 175 (97) | 362 (95) |
| 2 | 48 (0) | 20 (0) | 28 (0) | 19 (4) | 15 (5) | 15 (5) | 22 (5) | 6 (3) | 19 (5) |
| Age at diagnosis among individuals who develop cancer(s) (earliest age for those with multiple cancers) in years, median (IQR) | 70 (64–77) | 68 (62–76) | 72 (67–77) | 65 (59–70) | 72 (67–77) | 71 (64–78) | 71 (65–6) | 74 (66–79) | 74 (68–78) |
| Length of follow-up (longest follow-up for those with multiple cancers) in years, median (IQR) | 20 (18–22) | 20 (18–22) | 19 (17–21) | 11 (6–16) | 13 (9–17) | 13 (8–17) | 11 (7–16) | 14 (9–17) | 14 (10–17) |
| Dialect group (%) | | | | | | | | | |
| Hokkien | 10663 (49) | 6132 (51) | 4531 (47) | 260 (53) | 153 (50) | 185 (56) | 164 (40) | 95 (52) | 162 (43) |
| Cantonese | 11031 (51) | 5952 (49) | 5079 (53) | 235 (47) | 155 (50) | 147 (44) | 245 (60) | 86 (48) | 219 (57) |
| Highest education (%) | | | | | | | | | |
| No | 4629 (21) | 3878 (32) | 751 (8) | 128 (26) | 20 (6) | 123 (37) | 46 (11) | 85 (47) | 57 (15) |
| Primary level | 9760 (45) | 5082 (42) | 4678 (49) | 206 (42) | 146 (47) | 138 (42) | 232 (57) | 62 (34) | 228 (60) |
| Secondary or above | 7305 (34) | 3124 (26) | 4181 (44) | 161 (33) | 142 (46) | 71 (21) | 131 (32) | 34 (19) | 96 (25) |
| Body mass index in kg/m$^2$, median (IQR) | 23 (21–25) | 23 (21–25) | 23 (21–25) | 23 (21–25) | 23 (21–25) | 23 (21–24) | 23 (21–25) | 23 (20–24) | 23 (20–24) |
| Smoking status (%) | | | | | | | | | |
| Never | 15553 (72) | 11235 (93) | 4318 (45) | 472 (95) | 166 (54) | 296 (89) | 153 (37) | 129 (71) | 63 (17) |
| Ex-smoker | 2374 (11) | 261 (2) | 2113 (22) | 8 (2) | 66 (21) | 14 (4) | 108 (26) | 9 (5) | 74 (19) |
| Current smoker | 3767 (17) | 588 (5) | 3179 (33) | 15 (3) | 76 (25) | 22 (7) | 148 (36) | 43 (24) | 244 (64) |
| Number of cigarettes smoked (%) | | | | | | | | | |
| Does not smoke | 15553 (72) | 11235 (93) | 4318 (45) | 472 (95) | 166 (54) | 296 (89) | 153 (37) | 129 (71) | 63 (17) |
| <12 | 2408 (11) | 581 (5) | 1827 (19) | 14 (3) | 54 (18) | 26 (8) | 85 (21) | 36 (20) | 81 (21) |
| 13–22 | 2344 (11) | 206 (2) | 2138 (22) | 6 (1) | 53 (17) | 9 (3) | 108 (26) | 15 (8) | 135 (35) |
| ≥23 | 1389 (6) | 62 (1) | 1327 (14) | 3 (1) | 35 (11) | 1 (0) | 63 (15) | 1 (1) | 102 (27) |
| Alcohol consumption (%) | | | | | | | | | |
| Never/ occasionally | 19079 (88) | 11506 (95) | 7573 (79) | 470 (95) | 253 (82) | 315 (95) | 303 (74) | 174 (96) | 296 (78) |
| Weekly | 1885 (9) | 437 (4) | 1448 (15) | 20 (4) | 44 (14) | 10 (3) | 66 (16) | 5 (3) | 49 (13) |
| Daily | 730 (3) | 141 (1) | 589 (6) | 5 (1) | 11 (4) | 7 (2) | 40 (10) | 2 (1) | 36 (9) |
| Moderate physical activity (%) | | | | | | | | | |
| No | 16584 (76) | 9446 (78) | 7138 (74) | 380 (77) | 208 (68) | 269 (81) | 295 (72) | 143 (79) | 294 (77) |
| 1–3 hr/week | 3274 (15) | 1679 (14) | 1595 (17) | 69 (14) | 62 (20) | 43 (13) | 68 (17) | 23 (13) | 53 (14) |
| ≥ 3 hr/week | 1836 (8) | 959 (8) | 877 (9) | 46 (9) | 38 (12) | 20 (6) | 46 (11) | 15 (8) | 34 (9) |
| Vigorous physical activity/ strenuous sports at least once a week (%) | | | | | | | | | |
| No | 18467 (85) | 11221 (93) | 7246 (75) | 452 (91) | 239 (78) | 311 (94) | 342 (84) | 175 (97) | 314 (82) |

*Table 1 continued on next page*

*Table 1 continued*

| | Entire cohort | | | Individuals who developed cancer | | | | | |
|---|---|---|---|---|---|---|---|---|---|
| Yes | 3227 (15) | 863 (7) | 2364 (25) | 43 (9) | 69 (22) | 21 (6) | 67 (16) | 6 (3) | 67 (18) |
| Family history of any cancer in first-degree relatives (%) | | | | | | | | | |
| No | 18193 (84) | 10141 (84) | 8052 (84) | 404 (82) | 236 (77) | 281 (85) | 336 (82) | 165 (91) | 333 (87) |
| Yes | 3501 (16) | 1943 (16) | 1558 (16) | 91 (18) | 72 (23) | 51 (15) | 73 (18) | 16 (9) | 48 (13) |

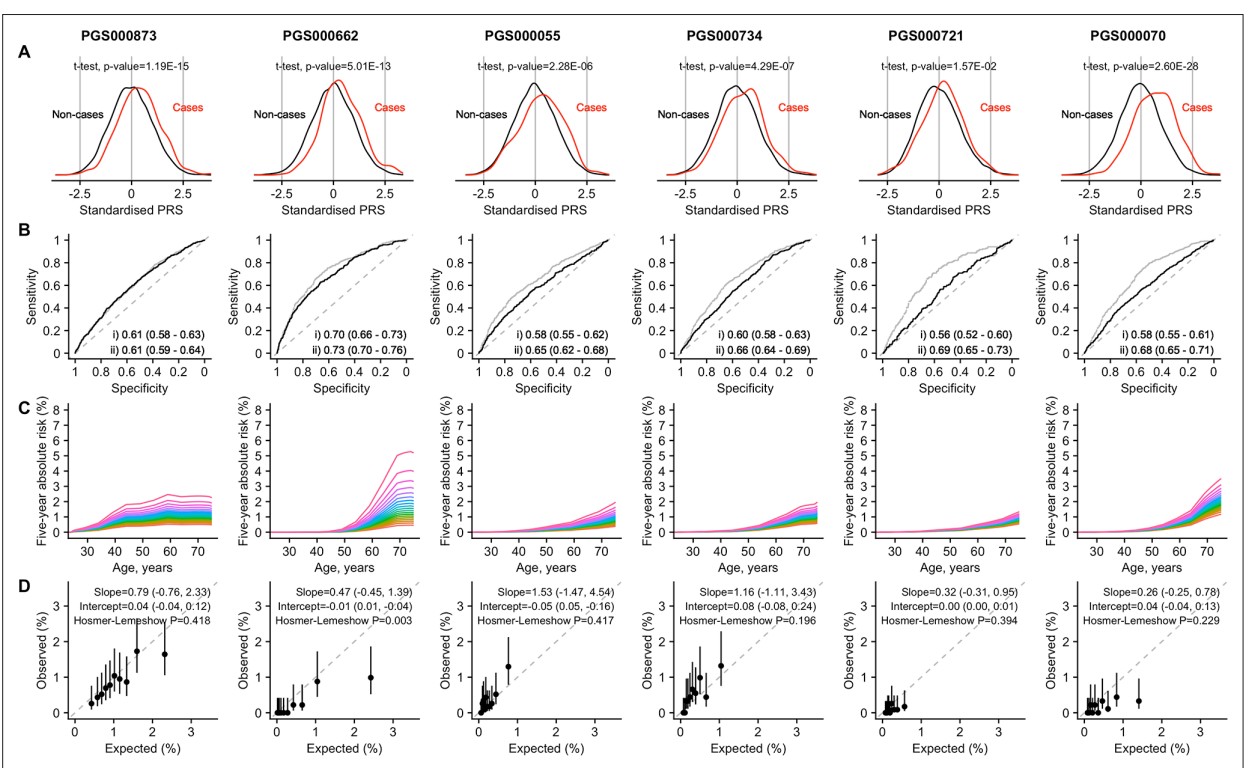

**Figure 1.** Site-specific polygenic risk scores (PRSs) performance assessment. (**A**) Distribution, (**B**) discrimination, (**C**) absolute risk association, and (**D**) calibration for each of the four common cancers studied (from left to right: breast, prostate, lung [female], lung [male], colorectal [female], and colorectal [male]). Two-sided, two-sample t-tests with a type I error of 0.05 were used to examine whether there was a difference in the distribution of standardised PRS (subtraction of mean value followed by the division by the standard deviation) between site-specific cancer cases and non-cancer controls (**A**). The PRSs showcased are the best-performing scores based on area under the receiver operator characteristic curve (AUC) values in the female and male populations, (**i**) unadjusted [solid line], and (ii) adjusted for age at recruitment [dashed line] (**B**). Each colored line in the plots for absolute risk association denotes a five percentile increase in the standardised PRS score in (**C**). Calibration calculated based on 5-year absolute risk by PRS deciles in (**D**). A prediction tool is considered more accurate when the AUC is larger. An AUC of 0.9–1.0 is considered excellent, 0.8–0.9 very good, 0.7–0.8 good, 0.6–0.7 sufficient, 0.5–0.6 bad, and less than 0.5 considered not useful (PMID: 27683318).

The online version of this article includes the following source data for figure 1:

**Source data 1.** Tables on absolute risk for breast cancer.

**Source data 2.** Tables on absolute risk for colorectal cancer.

**Source data 3.** Tables on absolute risk for lung cancer.

**Source data 4.** Tables on absolute risk for prostate cancer.

**Source data 5.** Tables on polygenic risk scores (PRS) performance assessment.

**Table 2.** Hazard ratios (HR) and corresponding 95% confidence intervals (CI) associated with polygenic risk score quintiles (Q) compared to the population median, using the Cox proportional hazards model and censored at 20 years after recruitment. Individuals were categorised into cancer-specific quintiles based on their cancer-specific polygenic risk score (PRS). All models were adjusted for age at recruitment.

| Cancer site – gender | Q1 | Q2 | Q3 | Q4 | Q5 |
|---|---|---|---|---|---|
| Breast – female | | | | | |
| Number of cases | 55 | 73 | 86 | 107 | 145 |
| HR (95% CI) | 0.61 (0.44–0.86) | 0.80 (0.59–1.09) | 1.00 (Referent) | 1.25 (0.94–1.66) | 1.64 (1.26–2.14) |
| Prostate – male | | | | | |
| Number of cases | 15 | 31 | 55 | 59 | 129 |
| HR (95% CI) | 0.28 (0.16–0.50) | 0.57 (0.37–0.88) | 1.00 (Referent) | 1.11 (0.77–1.60) | 2.52 (1.84–3.46) |
| Colorectal – female | | | | | |
| Number of cases | 47 | 43 | 53 | 66 | 101 |
| HR (95% CI) | 0.84 (0.57–1.25) | 0.80 (0.53–1.20) | 1.00 (Referent) | 1.27 (0.88–1.82) | 1.91 (1.37–2.67) |
| Colorectal – male | | | | | |
| Number of cases | 36 | 70 | 71 | 87 | 114 |
| HR (95% CI) | 0.51 (0.34–0.77) | 1.00 (0.72–1.39) | 1.00 (Referent) | 1.29 (0.94–1.76) | 1.67 (1.24–2.25) |
| Lung – female | | | | | |
| Number of cases | 25 | 26 | 41 | 36 | 40 |
| HR (95% CI) | 0.56 (0.34–0.92) | 0.55 (0.34–0.91) | 1.00 (Referent) | 0.89 (0.57–1.39) | 0.95 (0.61–1.47) |
| Lung – male | | | | | |
| Number of cases | 51 | 58 | 68 | 80 | 103 |
| HR (95% CI) | 0.72 (0.50–1.04) | 0.79 (0.56–1.13) | 1.00 (Referent) | 1.14 (0.82–1.57) | 1.46 (1.07–1.98) |

95% CI=0.58–0.63), female colorectal cancer (0.56 [0.52–0.60]), male lung cancer (0.58 [0.55–0.61]), and female lung cancer (0.55 [0.50–0.59]) (*Figure 1B*).

## Associations between PRS and the relative hazard of developing cancers

During the follow-up period of 20 years, the risk of acquiring breast, colorectal, or lung cancer increased significantly with higher PRS after adjusting for age at recruitment. Compared to the first PRS quintile, individuals in the highest quintile were more likely to develop the four cancers studied. The highest HR observed was for prostate cancer (8.99 [95%CI: 5.27–15.35]) and lowest for female lung cancer (1.69 [1.03–2.79]), adjusted for age at recruitment (*Supplementary file 1e*). Significant trends were found for the associations between PRS quintiles and site-specific cancers (p-trend ranges from $1.35 \times 10^{-25}$ for prostate cancer to 0.008 for female lung cancer, *Supplementary file 1e*).

Compared to the middle cancer-specific PRS quintile, individuals in the highest PRS quintile were 64% more likely to develop cancers of the breast, prostate, and colorectal (*Table 2*). Individuals in the lowest PRS quintile were associated with a 16–72% reduction in risk of developing these cancers. For female lung cancer, the PRS quintiles 1 and 2 were associated with 44% and 45% decreased risk compared to the middle quintile, respectively. However, the HR observed for PRS quintiles 4 (0.89 [0.57–1.39]) and 5 (0.95 [0.61–1.47]) were not significantly different when compared to the middle quintile.

Every SD increase in PRS is associated with 39–108% elevated risks of breast, prostate, and colorectal cancers ($p < 1.06 \times 10^{-8}$), adjusted for age, dialect group, education, BMI, smoking status,

**Table 3.** Associations between per standard deviation (SD) increase in site-specific polygenic risk scores and cancer occurrence. Hazard ratios (HR) and corresponding 95% confidence intervals (CI) were estimated using Cox proportional hazard models, adjusted for age at recruitment, dialect group, highest education attained, body mass index, smoking status, alcohol consumption, and physical activity. Follow-up time was censored at 20 years after recruitment. Significant results are shown in bold.

| | Cancer site | | | | | | | | | | | |
| --- | --- | --- | --- | --- | --- | --- | --- | --- | --- | --- | --- | --- |
| | Breast | | Prostate | | Colorectal – female | | Colorectal – male | | Lung – female | | Lung – male | |
| | HR (95% CI) | p-Value | HR (95% CI) | p-Value | HR (95% CI) | p-Value | HR (95% CI) | p-Value | HR (95% CI) | p-Value | HR (95% CI) | p-Value |
| Site-specific polygenic risk score, per SD increase | **1.47 (1.34–1.60)** | **5.80E-17** | **2.08 (1.85–2.34)** | **1.56E-33** | **1.39 (1.24–1.55)** | **1.06E-08** | **1.44 (1.30–1.59)** | **5.41E-12** | **1.21 (1.04–1.40)** | **1.10E-02** | **1.35 (1.22–1.49)** | **1.01E-08** |
| Age at recruitment, years | 1.00 (0.99–1.02) | 5.82E-01 | **1.09 (1.07–1.10)** | **6.34E-23** | **1.07 (1.05–1.09)** | **7.24E-17** | **1.06 (1.05–1.08)** | **9.53E-18** | **1.07 (1.05–1.10)** | **1.65E-10** | **1.09 (1.07–1.10)** | **1.46E-27** |
| Dialect group (Cantonese vs Hokkien) | 0.88 (0.73–1.05) | 1.61E-01 | 0.98 (0.78–1.24) | 8.86E-01 | 0.78 (0.62–0.99) | 3.96E-02 | 1.22 (0.99–1.50) | 6.78E-02 | 0.92 (0.67–1.25) | 5.78E-01 | 1.07 (0.87–1.33) | 5.21E-01 |
| Highest education (primary vs no) | 1.21 (0.95–1.53) | 1.20E-01 | 1.32 (0.81–2.14) | 2.65E-01 | 1.08 (0.83–1.41) | 5.60E-01 | 0.98 (0.70–1.37) | 8.91E-01 | 0.83 (0.58–1.19) | 3.11E-01 | 0.87 (0.64–1.18) | 3.67E-01 |
| Highest education (secondary or above vs no) | **1.54 (1.18–2.01)** | **1.57E-03** | 1.60 (0.98–2.63) | 6.17E-02 | 1.06 (0.76–1.48) | 7.46E-01 | 0.80 (0.55–1.16) | 2.33E-01 | 1.10 (0.69–1.74) | 6.87E-01 | **0.63 (0.44–0.90)** | **1.16E-02** |
| Body mass index, kg/m² | **1.04 (1.02–1.07)** | **1.28E-03** | 1.01 (0.98–1.05) | 5.15E-01 | 0.99 (0.96–1.02) | 5.33E-01 | 1.02 (0.98–1.05) | 3.19E-01 | 0.97 (0.92–1.01) | 1.58E-01 | 0.97 (0.93–1.00) | 5.60E-02 |
| Smoking status (ex-smoker vs non-smoker) | 0.90 (0.45–1.83) | 7.81E-01 | **0.68 (0.50–0.92)** | **1.32E-02** | 1.51 (0.86–2.66) | 1.55E-01 | 1.17 (0.90–1.52) | 2.36E-01 | **2.16 (1.04–4.48)** | **3.86E-02** | **1.99 (1.41–2.83)** | **1.09E-04** |
| Smoking status (current smoker vs non-smoker) | 0.83 (0.49–1.39) | 4.72E-01 | **0.70 (0.52–0.93)** | **1.52E-02** | 1.10 (0.69–1.75) | 6.85E-01 | 1.22 (0.96–1.56) | 1.08E-01 | **5.78 (3.98–8.38)** | **2.69E-20** | **5.15 (3.83–6.91)** | **1.17E-27** |
| Alcohol consumption (weekly vs never/ occasionally) | 1.04 (0.65–1.67) | 8.74E-01 | 0.98 (0.70–1.39) | 9.29E-01 | 0.76 (0.38–1.54) | 4.46E-01 | 1.31 (1.00–1.73) | 5.39E-02 | 0.72 (0.27–1.96) | 5.23E-01 | 0.89 (0.65–1.22) | 4.81E-01 |
| Alcohol consumption (daily vs never/ occasionally) | 0.71 (0.27–1.91) | 5.00E-01 | 0.74 (0.40–1.36) | 3.32E-01 | 1.55 (0.69–3.49) | 2.89E-01 | **1.64 (1.15–2.34)** | **6.54E-03** | 0.66 (0.16–2.68) | 5.63E-01 | 1.21 (0.85–1.73) | 2.81E-01 |
| Moderate physical activity (1–3 hr/week vs no) | 0.98 (0.75–1.28) | 8.80E-01 | 1.17 (0.87–1.57) | 2.97E-01 | 0.88 (0.63–1.24) | 4.74E-01 | 1.02 (0.78–1.35) | 8.79E-01 | 0.99 (0.62–1.58) | 9.73E-01 | 0.90 (0.67–1.23) | 5.17E-01 |
| Moderate physical activity (≥3 hr/week vs no) | 1.17 (0.86–1.60) | 3.20E-01 | 0.99 (0.68–1.45) | 9.78E-01 | 0.59 (0.37–0.96) | 3.33E-02 | 1.10 (0.80–1.52) | 5.45E-01 | 0.96 (0.54–1.70) | 8.78E-01 | 0.86 (0.59–1.26) | 4.36E-01 |
| Vigorous physical activity/ strenuous sports at least once a week (yes vs no) | 1.24 (0.90–1.70) | 1.89E-01 | 1.05 (0.79–1.41) | 7.16E-01 | 1.09 (0.68–1.74) | 7.25E-01 | 0.75 (0.57–1.00) | 5.06E-02 | 0.58 (0.24–1.42) | 2.30E-01 | 0.95 (0.72–1.26) | 7.37E-01 |
| Family history (yes vs no) | 1.14 (0.90–1.45) | 2.67E-01 | **1.53 (1.16–2.02)** | **2.47E-03** | 1.08 (0.79–1.48) | 6.20E-01 | 1.24 (0.95–1.62) | 1.09E-01 | 0.67 (0.40–1.13) | 1.33E-01 | 0.97 (0.71–1.33) | 8.63E-01 |

physical activity, alcohol consumption, and family history (*Table 3*). The increased risk for female and male lung cancer was lower than the other three cancers (HR$_{female}$: 1.21 [1.04–1.40], p=0.011; HR$_{male}$: 1.35 [1.22–1.49], p=1.01 × 10$^{-8}$). Age at recruitment is significantly associated with elevated risks of developing all cancers, with the exception of female breast cancer (HR: 1.00 [0.99–1.02], p=0.571). Highest education level and BMI were positively correlated with breast cancer risk. Smoking was significantly associated with an ~30% reduction in risk of prostate cancer but increased the risk of lung cancer by approximately two- and fivefold for past and current smokers, compared to non-smokers, respectively. Alcohol consumption increased the risk of both female and male colorectal cancer by

approximately 60% but was only significant for male colorectal cancer. Family history of cancer was only significantly associated with an increased risk for prostate cancer (HR: 1.53 [1.16–2.02], p=7.59 × 10⁻⁴).

All Cox models presented in *Tables 2 and 3* did not violate the proportionality assumption for the PRS studied (p-values of *cox-zph*() for PRS were >0.05).

## Association of PRS with absolute risk

In terms of the 5-year absolute risk of developing site-specific cancers, the largest difference between the highest and lowest PRS categories was observed for prostate cancer, followed by breast cancer (*Figure 1C*). A separation of the absolute risk curves was observed for female breast cancer already at age 30 years. For prostate cancer, the separation of curves was observed only after age 50 years. Slight separation of the curves began after 50 years of age for colorectal and lung cancer.

## PRS calibration

In general, predicted risks for the higher PRS categories did not correspond well to the observed proportions for female breast, male prostate, and female lung cancers (*Figure 1D*); in particular, predicted risks were overestimated for the higher risk categories. Overestimation of risk was observed for all PRS categories for male lung cancer. In contrast, predicted risks were underestimated for both female and male colorectal cancers. Nonetheless, the CI associated with the calibration slopes for all cancers included 1, with the exception of female (0.32 [-0.31 to 0.95]) and male lung cancers (0.26 [-0.25, 0.78]).

## Discussion

Precision prevention in oncology is based on the idea that an individual's risk, which is influenced by genetics, environment, and lifestyle factors, is linked to the amount of benefit achieved through cancer screening (*Roberts, 2018*). Risk stratification for cancer screening can be used in this framework to identify and recommend screening for persons with a high enough cancer risk that the benefits outweigh the risks. Several PRS prediction models have been established for site-specific cancers, each with its own set of strengths and limitations, and different risk models may produce different results for the same individual.

In an increasingly inclusive world, genetic studies fall short on diversity. According to a 2009 study, an overwhelming 96% of people who took part in genome-wide association studies (GWAS) were of European ancestry (*Need and Goldstein, 2009*). GWAS results are the backbone on which PRS is developed. A concern raised was that, without representation from a broader spectrum of populations, genomic medicine may be limited to benefitting 'a privileged few' (*Popejoy and Fullerton, 2016*).

Genetic studies in 2016 showed that the proportion of people not of European ancestry included in GWAS has increased to approximately 20% (*Popejoy and Fullerton, 2016*). Most of this rise can be attributed to more research on Asian ancestry communities in Asia (*Popejoy and Fullerton, 2016*). With increasing interest worldwide in using a risk-based approach to screening programs over the current age-based paradigm, this progress raises questions on whether selected established PRS shown to perform well in European-based populations has equal utility in Asians. Nonetheless, as our results show, most of the populations from which PRS were developed are still predominantly of European ancestry.

In accordance with published Polygenic Risk Score Reporting Standards, we reported PRS distribution, discrimination, absolute risk association, and calibration for each of the four common cancers studied (*Wand et al., 2021*). Our results show that cancer cases were associated with higher PRS compared to non-cancer controls. In the age-adjusted models, a constant trend between PRS percentile rank and observed cancer risk in our study population supports the validity of PRS for breast, prostate, and colorectal cancers, but not for lung cancer. The best-performing PRS for female breast cancer was able to stratify women into distinct bands of breast cancer risk at an earlier age, and across all ages, suggesting that it could be a useful prediction tool in risk-based breast cancer screening in combination with other risk factors specific to breast cancer (*Ho et al., 2022*). This PRS has been incorporated into a pilot risk-based breast cancer screening study in a comparable study population (*Liu*

*et al., 2022*). The best-performing PRS for prostate and male colorectal cancers in this study appeared to exhibit sufficient discriminatory ability and predictive value, especially for older participants.

PRS may be of limited use in predicting female colorectal and female/male lung cancer. The least predictive value was in lung cancer, which could be related to the higher prevalence of EGFR mutant lung cancer which has an Asian predilection, thus less amenable to PRS developed in Caucasian population (*Shigematsu et al., 2005*). It is reassuring to see tobacco smoking is a strong risk factor for lung cancer in our dataset. However, smoking appeared to be associated with a protective effect for prostate cancer. While smoking is a well-known risk factor for many cancers (*Jacob et al., 2018*), in particular lung cancer, observational studies frequently show that smokers are associated with a lower incidence of prostate cancer (*Rohrmann et al., 2013*; *Watters et al., 2009*; *Adami et al., 1996*; *Lund Nilsen et al., 2000*; *Engeland et al., 1996*; *Islami et al., 2014*; *Ordóñez-Mena et al., 2016*; *Giovannucci et al., 2007*). However, a Mendelian randomisation study did not support the association (*Larsson et al., 2020*).

There is room for improvement in the discriminatory ability of PRS (*Lewis and Green, 2021*). As noted by Lambert et al. in a review, a wider divergence between the average scores of cases and non-cases (quantified by AUC) and associated effect sizes (odds ratio and SD) is expected when PRS explains more of the heredity for each trait (*Lambert et al., 2019*). Larger GWAS sample sizes of appropriate ancestries and the inclusion of rarer genetic variants, obtained through other methods such as whole-genome sequencing, would likely be required to boost explained heritability (*Lambert et al., 2019*). In addition, group-wise estimates, which arbitrarily classify the top 10%, 5%, or 1% of samples as the at-risk group, are not optimal for decisions at the individual level (*Lewis and Green, 2021*). Emerging new methodologies that estimate probability values for hypothetically assigning an individual as at risk or not at risk, thus providing individuals with more clarity, may help to overcome this limitation (*Sun et al., 2021*). At this point, PRS may not have yet reached the standards as a clinical tool by itself. However, it is still helpful in guiding screening decisions and supplementing established protocols (*Polygenic Risk Score Task Force of the International Common Disease Alliance, 2021*).

As highlighted by Wei et al., the reliability of score values is necessary for application at the individual level (*Wei et al., 2022*). Even when a PRS has adequate discrimination, estimated risks can be unreliable (*Van Calster et al., 2019*). Our results show that cancer risk estimates based on PRS developed using populations of European ancestry are not optimally calibrated for our Asian study population. However, the CI associated with the calibration slopes for all cancers except for female and male lung included the value of 1, suggesting that the overall calibration is not poor. For female and male lung cancers associated with low AUC values, poor calibration is not unexpected. All PRSs except PGS000662 (prostate cancer) passed the formal Hosmer-Lemeshow goodness-of-fit test. Males in the first 5 deciles of PGS000662 did not develop prostate cancer, suggesting that a linear fit may not be appropriate. A hard threshold beginning from the 6th decile may perform better at identifying males at elevated risk of developing prostate cancer.

Poorly calibrated PRS can be misleading and have clinical repercussions (*Van Calster et al., 2019*; *Van Calster and Vickers, 2015*). Underestimation of risk may result in a false sense of security. Over-estimation of risk may cause unnecessary anxiety, misguided interventions, and overtreatment. In a population-wide screening setting, however, where the return of PRS results can be designed such that only high-risk individuals are highlighted, underestimation of risk may be less of an issue. Arguably, with parallel input from other risk factors and evaluation by healthcare specialists, the over-estimation of risk that results in a higher number of at-risk individuals identified may increase the number of cancers potentially detected early. Nonetheless, suitable correction factors will be required to ensure the reliability of PRS prior to clinical implementation.

While the study population used in this analysis comprises less than a thousand cases of the most common cancers examined, the SCHS, established between April 1993 and December 1998, is one of the largest population-based Asian cohorts in the world with high-quality prospective data on exposure and comprehensive capture of morbidity and mortality. All cancer cases are incident cases diagnosed over three decades of follow-up. Blood samples were collected from a subset of SCHS participants who were alive and contactable between 1999 and 2004 (after the recruitment period 1993–1998). While we attempted to adjust associations between PRSs and incident cancers in the study by including multiple related risk factors as covariates in Cox proportional hazards models, we acknowledge the potential of survival biases in the study. This is one of the best resources to evaluate

the utility of PRS in a prospective manner. The findings open a window in our current understanding of which PRS is relevant and ready to be deployed in risk-based cancer screening studies. Nonetheless, it should be noted that among the PRS interrogated from the PGS Catalog, the 'best' PRS selected in this study may be only superficially superior in terms of AUC (i.e. trailing decimal places) over the 'next best' PRS. We further tested the sensitivity of the PRS selection using a time-to-event metric for AUC (at 5 years), and the differences found were non-informative from the logistic regression (*Supplementary file 1f*). To increase the number of events, we combined the males and females for lung and colorectal cancers. The resulting AUCs (from the logistic regression) were not appreciably different from the sex-specific analysis (*Supplementary file 1g*).

Ethnic representation in PRS model development, PRS validation, limited discriminative ability in the general population, ill calibration, insufficient healthcare professional and patient education, and healthcare system integration are all hurdles that must be crossed before PRS can be implemented responsibly as a public health instrument (*Lewis and Vassos, 2020*; *Slunecka et al., 2021*). While nationwide screening programs have helped to raise cancer awareness, there is still a need to improve the effectiveness and efficiency of cancer screening in Asian countries such as Singapore, given the steadily rising incidence rates. Despite the challenges, a risk-based screening strategy that includes the use of PRS should be actively examined for research and implementation.

## Acknowledgements

We thank the Singapore Cancer Registry for the identification of incident cancer cases among participants of the SCHS and Siew-Hong Low of the National University of Singapore for supervising the fieldwork of the SCHS.

## Additional information

### Competing interests

Rajkumar Dorajoo: received a grant from the Agency for Science, Technology and Research Career Development Award (A*STAR CDA - 202D8090), and from Ministry of Health Healthy Longevity Catalyst Award (HLCA20Jan-0022). The author has no other competing interests to declare. The other authors declare that no competing interests exist.

### Funding

| Funder | Grant reference number | Author |
| --- | --- | --- |
| National Research Foundation Singapore | PRECISE | Jingmei Li |
| National Medical Research Council | NMRC/CSA/0055/2013 | Woon-Puay Koh |
| National Research Foundation Singapore | NRF-NRFI2018-01 | Chiea Chuen Khor |
| Ministry of Health -Singapore | HLCA20Jan-0022 | Rajkumar Dorajoo |
| National Institutes of Health | R01 CA144034 | Jian-Min Yuan |
| National Institutes of Health | UM1 CA182876 | Jian-Min Yuan |

The funders had no role in study design, data collection and interpretation, or the decision to submit the work for publication.

### Author contributions

Peh Joo Ho, Conceptualization, Formal analysis, Visualization, Methodology, Writing – original draft, Writing – review and editing; Iain BeeHuat Tan, Writing – review and editing, Interpretation of data; Dawn Qingqing Chong, Writing – review and editing; Chiea Chuen Khor, Data curation, Funding

acquisition, Writing – review and editing; Jian-Min Yuan, Woon-Puay Koh, Resources, Data curation, Funding acquisition, Writing – review and editing; Rajkumar Dorajoo, Conceptualization, Resources, Data curation, Funding acquisition, Methodology, Writing – original draft, Writing – review and editing; Jingmei Li, Conceptualization, Resources, Data curation, Supervision, Funding acquisition, Methodology, Writing – original draft, Writing – review and editing

**Author ORCIDs**
Peh Joo Ho ⓘD http://orcid.org/0000-0002-3017-4023
Jingmei Li ⓘD http://orcid.org/0000-0001-8587-7511

**Ethics**
Human subjects: The study was approved by the Institutional Review Boards of the National University of Singapore, University of Pittsburgh, and the Agency for Science, Technology and Research (A*STAR, reference number 2022-042). Written, informed consent was obtained from all study participants.

**Decision letter and Author response**
Decision letter https://doi.org/10.7554/eLife.82608.sa1
Author response https://doi.org/10.7554/eLife.82608.sa2

---

## Additional files

**Supplementary files**
• Supplementary file 1. Supplementary files a-g, presenting supplementary figure and tables.
• MDAR checklist
• Source code 1. R codes on the statistical analysis.

**Data availability**
All polygenic risk scores used in this study are publicly available in the PGS Catalog (https://www.pgscatalog.org; *Lambert et al., 2021*). The data that support the findings of our study are available from the corresponding authors of the study upon reasonable request (Dr Rajkumar s/o Dorajoo, dorajoor@gis.a-star.edu.sg and Dr Jingmei Li, lijm1@gis.a-star.edu.sg). More information regarding the data access to SCHS can be found at: https://sph.nus.edu.sg/research/cohort-schs/. The data are not publicly available due to Singapore laws. Figure 1—source data 1 contains the numerical data used to generate Figure 1. The code for the study is uploaded as Source code 1.

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
