## [Editor Report]

Utilizing one of the largest prospective Asian cohorts with long-term follow-up data, the important study reveals the utility of polygenic risk scores in identifying high-risk individuals from specific cancers. The convincing results reached by the high-quality data highlight the translational significance of this study, which may be important for future exploration in the field of cancer epidemiology.

---

## [Decision Letter]

**Decision letter after peer review:**

Thank you for submitting your article "Polygenic risk scores for the prediction of common cancers in East Asians: A population-based prospective cohort study" for consideration by *eLife*. Your article has been reviewed by 2 peer reviewers, and the evaluation has been overseen by a Reviewing Editor and Caigang Liu as the Senior Editor. The reviewers have opted to remain anonymous.

The reviewers have discussed their reviews with one another, and revisions are necessary for the manuscript. For your guidance, reviewers' comments are appended below. If you decide to revise your manuscript, please revise your work guided by the reviewers' suggestions, and provide a point-by-point response to the following suggestions and concerns.

*Reviewer #1 (Recommendations for the authors):*

This is an important paper that has the potential to contribute to our understanding of how to take forward polygenic risk scores developed primarily based on individuals of European descent on the large Asian population.

1. However, by focusing on a combined analysis of 4 cancers, the main findings appear to have been lost.

a. The authors note in the discussion that the PRSes developed in individuals of European descent has not well calibrated for the Asian population One would not expect PRS with poor AUC to be well calibrated, hence poor calibration for lung cancer-PRS may be expected in this study. For other cancer-PRSs, the confidence interval of the calibration slopes indicating that they are not significantly different from one, hence no strong suggestion of poor calibration overall, except maybe for prostate PRS. Would it be possible that observed prostate cancer incidence is underestimated due to survival bias?

b. For each of the 4 cancers, it would be helpful to report features associated with translatability of PRS from Europeans to Asians. The authors report that "no significant association (P>0.05) was found between number of variants included in the various PRS evaluated for each cancer and discriminatory ability" – an exploratory analysis of other features associated with performance would be helpful.

c. In Table 2, the authors appear to have grouped individuals into quintiles based on a combined score. It is not clear what is the intent for this analysis – is the aspiration to roll out a risk stratified screening strategy based on "high risk for any of 4 cancers" and then to screen for all 4 cancers? The corresponding text in Line 315-330 needs clarification, it is not clear what is the intent for repeating the analyses with different PRS quintiles as reference group.

d. In Table 3, the authors reported the association of PRS together with other demographic/risk factors. It hasn't been made clear the reason for such exhaustive adjustment in the model. If the intent is to quantify the attenuated amount in PRS association after accounting for potential confounders, then unadjusted PRS would need to be presented and the point needs to be made clear in the text.

e. In Table 4, the authors intend to quantify the proportion of cases captured by the at-risk groups as defined by PRSs. However, what's the rational of the HR cutoffs? Why not use the absolute risks in Figure 1(c) instead? What is the intent to include other risk factors? Higher proportion of cases captured would be expected with additional risk factors, however, it may be an overestimation here given that the RRs of other risk factors were estimated from the same cohort?

2. Have the authors sufficiently addressed survival bias? The cohort of 68k individuals was established in 1993-1998, but blood samples from 28k individuals were only collected in 1999-2004, and of these, 21,694 were analysable. Was the Singapore Cancer Registry data systematically collected from 1993?

*Reviewer #2 (Recommendations for the authors):*

1. My primary concern is lack of clarity in why not all cancer PRS were carried forward from Supplemental Table 1 to evaluation in Supplemental Table 2. The PRS from the largest prostate cancer GWAS to date (Conti 2021), including 12% Asian individuals, is represented in Supplemental Table 1 (PGS ID: PGS000662) but is not carried forward for evaluation in Supplemental Table 2. Similarly, a lung cancer PRS developed specifically for a Chinese population (Dai 2020) is also represented in Supplemental Table 1 (PGS ID: PGS000070) but not carried forward for evaluation. I have not performed similar inspection for the breast and colorectal cancer PRS, but this raises the concern about incomplete evaluation of available (and even best) PRS.

2. It is not clear that AUC should be used to identify the "best" PRS for each cancer in this population, particularly since from what I can tell the author did not use a time-dependent AUC method and excluded all baseline cases from analyses. A time-to-event metric would seen to be a more appropriate way to identify the "best" PRS in these analyses, either using an AUC method for incident event data or using risk association.

3. Although the term "predictive ability" is used in the landmark Wand 2021 paper (ref 31), I recommend using more specific terms in this manuscript, starting with the abstract (line 66). I think the correct terms would be "relative risk association" and "absolute risk association," as appropriate. Still, Ref 31 (Wand 2021) is helpful and should be introduced in the Introduction to orient to reader to the reporting framework the authors use in presenting this work.

4. Can the authors justify why they performed only sex-stratified analyses for lung and colorectal cancer? The GWAS for these conditions included both sexes. Given the relatively small number of cases, it would be worth performing all analyses in the cohort overall. Sex-stratified analyses could be presented in the supplemental materials, if desired.

5. Lines 182-184: How is it known that only <1% of participants migrated out of Singapore? Are other data available to confirm their ongoing residence in Singapore and absence of cancer diagnosis? E.g., other medical diagnoses or other data?

6. It is reassuring to see cigarette smoking as a strong lung cancer risk factor, but the apparent protective effect of smoking on prostate cancer contradicts current evidence. The authors should discuss this in the Discussion.

7. Lines 274-292 could be shortened, simply referring the reader to most of these results in Table 1.

8. Lines 368-387: This section of the results seems out of order compared with the Methods. I recommend reorganizing the Results as follows: "Characteristics of the study population," "Lack of Asian representation in PRS development," "PRS discriminatory ability," "PRS distribution," "Associations between PRS and relative hazard of developing cancers," "Number of cancers that developed within PRS at-risk groups," "Association of PRS with absolute risk," "PRS calibration."

9. Calibration should be formally tested (e.g., with a Hosmer-Lemeshow or more sophisticated test) in addition to the visual inspection presented.

10. Lines 470-476 of the Discussion stray a little from the focused thesis of this work.

[Editors' note: further revisions were suggested prior to acceptance, as described below.]

Thank you for resubmitting your work entitled "Polygenic risk scores for the prediction of common cancers in East Asians: A population-based prospective cohort study" for further consideration by *eLife*. Your revised article has been evaluated by Caigang Liu (Senior Editor) and a Reviewing Editor.

The manuscript has been improved but there are some remaining issues that need to be addressed, as outlined below:

1. Your response to the initial critique #1 is valid. The response itself contains errors (e.g. incorrect PGS numbers given for specific disease PRS), but the supplemental tables appear correct. The caption for Figure 1 is missing, which should explain which column corresponds to which disease and in which sex.

2. Please add the methods of your new time-to-event AUC sensitivity analysis to the main Methods of the manuscript, including a reference to the R package used. Also please state explicitly in the Methods your approach to choosing the best PRS for each disease (or disease-sex combination): logistic regression AUC.

3. Please explicitly state this rationale for sex-stratified analyses for these cancers in the Methods.

---

## [Author Response]

Reviewer #1 (Recommendations for the authors):This is an important paper that has the potential to contribute to our understanding of how to take forward polygenic risk scores developed primarily based on individuals of European descent on the large Asian population.1. However, by focusing on a combined analysis of 4 cancers, the main findings appear to have been lost.a. The authors note in the discussion that the PRSes developed in individuals of European descent has not well calibrated for the Asian population One would not expect PRS with poor AUC to be well calibrated, hence poor calibration for lung cancer-PRS may be expected in this study. For other cancer-PRSs, the confidence interval of the calibration slopes indicating that they are not significantly different from one, hence no strong suggestion of poor calibration overall, except maybe for prostate PRS. Would it be possible that observed prostate cancer incidence is underestimated due to survival bias?

We have added the point that the reviewer highlighted about confidence intervals associated with the calibration slopes in the Results:

In Results (section “PRS calibration”): “In general, predicted risks for the higher PRS categories did not correspond well to the observed proportions for female breast, male prostate, and female lung cancers (Figure 1D); in particular, predicted risks were overestimated for the higher risk categories. Overestimation of risk was observed for all PRS categories for male lung cancer. In contrast, predicted risks were underestimated for both female and male colorectal cancers. Nonetheless, the confidence intervals associated with the calibration slopes for all cancers included 1, with the exception of female (0.32 [-0.31 to 0.95]) and male lung cancers (0.26 [-0.25, 0.78]).”

The prostate cancer PRS has poor evidence of a good fit.

In the Discussion paragraph 5, we have added (in bold):

“Our results show that cancer risk estimates based on PRS developed using populations of European ancestry are not optimally calibrated for our Asian study population. However, the confidence intervals associated with the calibration slopes for all cancers except for female and male lung included the value of 1, suggesting that the overall calibration is not poor. For female and male lung cancers associated with low AUC values, poor calibration is not unexpected. All PRSs except PGS000662 (prostate cancer) passed the formal Hosmer-Lemeshow goodness-of-fit test. Males in the first 5 deciles of PGS000662 did not develop prostate cancer, suggesting that a linear fit may not be appropriate. A hard threshold beginning from the 6th decile may perform better at identifying males at elevated risk of developing prostate cancer.”

b. For each of the 4 cancers, it would be helpful to report features associated with translatability of PRS from Europeans to Asians. The authors report that "no significant association (P>0.05) was found between number of variants included in the various PRS evaluated for each cancer and discriminatory ability" – an exploratory analysis of other features associated with performance would be helpful.

We did not find evidence of association between the number of variants and the calibration (expected/observed or Hosmer-Lemeshow p-value) of the PRS in our study population. We summarise the results from the additional analysis here in Author response table 1 and ( (Author response image 1) ) .

**Author response table 1. sa2table1:** Linear associations between features performance (AUC, calibration [expected/ observed], and Hosmer-Lemeshow p-values) and the number of variants in the polygenic risk score, by cancer type.

**Cancer site – gender**	**Feature**	**Linear association p-value**	**Max value of feature**
Breast – Female	AUC	P=0.864	0.61075098
Breast – Female	Calibration (E/O)	P=0.748	1.50854654
Breast – Female	Hosmer-Lemeshow p-value	P=0.847	0.960221
Prostate – Male	AUC	P=0.403	0.72849342
Prostate – Male	Calibration (E/O)	P=0.567	4.74364593
Prostate – Male	Hosmer-Lemeshow p-value	P=0.844	0.4708587
Colorectal – Female	AUC	P=0.734	0.64886163
Colorectal – Female	Calibration (E/O)	P=0.779	1.00833383
Colorectal – Female	Hosmer-Lemeshow p-value	P=0.296	0.9056375
Colorectal – Male	AUC	P=0.666	0.66361296
Colorectal – Male	Calibration (E/O)	P=0.752	1.03242402
Colorectal – Male	Hosmer-Lemeshow p-value	P=0.047	0.8180789
Lung – Female	AUC	P=0.728	0.68602239
Lung – Female	Calibration (E/O)	P=0.869	2.97417751
Lung – Female	Hosmer-Lemeshow p-value	P=0.111	0.8170288
Lung – Male	AUC	P=0.451	0.68032583
Lung – Male	Calibration (E/O)	P=0.385	2.78560993
Lung – Male	Hosmer-Lemeshow p-value	P=0.404	0.9451139

**Author response image 1. sa2fig1:** Distributions of AUC, calibration (expected/ observed), and Hosmer-Lemeshow p-values for features of performance, by cancer type.

c. In Table 2, the authors appear to have grouped individuals into quintiles based on a combined score. It is not clear what is the intent for this analysis – is the aspiration to roll out a risk stratified screening strategy based on "high risk for any of 4 cancers" and then to screen for all 4 cancers? The corresponding text in Line 315-330 needs clarification, it is not clear what is the intent for repeating the analyses with different PRS quintiles as reference group.

Cancer-specific quintiles were used for each cancer-specific analysis. The premise of this analysis is to identify the best-performing PRS for each cancer for the purpose of informing screening strategies for each respective cancer. To clarify, we have made changes to the text as follows:

In Methods (section “Associations between PRS and risk of developing cancers”): “The associations between cancer-specific PRS quintiles (where individuals ranked by PRS were categorized into quintiles, using the middle quintile [40 to 60%] as a reference to reflect the average risk of the population) and the incidence of site-specific cancers were investigated using Cox proportional hazards modeling to estimate hazard ratios (HR) and corresponding 95% confidence intervals (CI), using time since recruitment as the time scale, and adjusted for age at recruitment.”

In Results (section “Associations between PRS and the relative hazard of developing cancers”): “Compared to the middle cancer-specific PRS quintile, individuals in the highest PRS quintile were 64% more likely to develop cancers of the breast, prostate, and colorectal (Table 2).”

Table 2 caption: “Table 2. Hazard ratios (HR) and corresponding 95% confidence intervals (CI) associated with polygenic risk score quintiles (Q) compared to the population median, using the Cox proportional hazards model and censored at 20 years after recruitment. Individuals were categorized into cancer-specific quintiles based on their cancer-specific PRS. All models were adjusted for age at recruitment.”

d. In Table 3, the authors reported the association of PRS together with other demographic/risk factors. It hasn't been made clear the reason for such exhaustive adjustment in the model. If the intent is to quantify the attenuated amount in PRS association after accounting for potential confounders, then unadjusted PRS would need to be presented and the point needs to be made clear in the text.

We have clarified in Methods (section “Associations between PRS and risk of developing cancers”): “PRS is known to have “portability” issues related to genetic ancestry and demographics (10.1038/s41588-019-0379-x, https://elifesciences.org/articles/48376). Hence, we adjusted for variables in the models, including age at recruitment, dialect group (Hokkien or Cantonese), highest level of education (no formal education, primary school, or secondary or higher), body mass index (continuous, kg/m2), cigarette smoking (non-smoker, ex-smoker, current smoker), alcohol consumption (never, weekly, daily), moderate physical activity (none, 1-3h/week, ≥3h/week), vigorous work/strenuous physical activity at least once a week (no or yes), and familial history of cancer (no or yes).”

Results for the unadjusted PRS analyses have been added to Supplementary file 1d, columns N to Q.

e. In Table 4, the authors intend to quantify the proportion of cases captured by the at-risk groups as defined by PRSs. However, what's the rational of the HR cutoffs? Why not use the absolute risks in Figure 1(c) instead? What is the intent to include other risk factors? Higher proportion of cases captured would be expected with additional risk factors, however, it may be an overestimation here given that the RRs of other risk factors were estimated from the same cohort?

To minimize confusion, we have removed the results from Table 4. Using the absolute risk as the cut-off will not increase information known about the group identified as high-risk. The proportion of individuals identified as at high risk will be greater than the percentile identified by the absolute risk cut-off (x%). In addition, if the PRS is well calibrated, we expect >x% of these high-risk individuals to develop the disease.

2. Have the authors sufficiently addressed survival bias? The cohort of 68k individuals was established in 1993-1998, but blood samples from 28k individuals were only collected in 1999-2004, and of these, 21,694 were analysable. Was the Singapore Cancer Registry data systematically collected from 1993?

We thank the reviewer for highlighting this point. First, we agree that survival bias could exist since, we could only include participants who survived to participate in the follow-up interview and who also agreed to give blood samples for research. For example, in the breast cancer evaluation, those who gave blood samples were younger (mean age of 55.1 years versus 57.1 years at recruitment) and were also more likely to have received education (32.9 percent with no formal education versus 45.4 percent) compared to women who did not give blood. As such, Cox proportional hazards models evaluating associations between PRS and incident cancers in the study were additionally adjusted for these related risk factors collected at recruitment, including age at recruitment, highest level of education, body mass index, cigarette smoking status, alcohol consumption, physical activity and familial history of cancer. In the revised version of the manuscript, we have further included survival bias as a potential study limitation:

In Discussion (second last paragraph): “Blood samples were collected from a subset of SCHS participants who were alive and contactable between 1999 and 2004 (after the recruitment period 1993 – 1998). While we attempted to adjust associations between PRSs and incident cancers in the study by including multiple related risk factors as covariates in Cox proportional hazards models, we acknowledge the potential of survival biases in the study.”

We have added more information on the Singapore Cancer Registry:

In Methods (section “Selection of common cancers”): “Identification of incident cases of cancer was accomplished by record linkage of all surviving cohort participants with the database of the nationwide Singapore Cancer Registry [20]. The Singapore Cancer Register was founded in 1968. Prior to 2009, reporting of neoplasms by all medical practitioners and pathology laboratories to the registry is voluntary (10.1016/j.canep.2016.06.006). The registry's staff compares cancer patient hospital discharges and death certificates to registered cases for verification. Completeness of reporting in the 1970s is 96% and in the 1990s, it was close to 100% (10.1186/1471-2407-6-261).”

Reviewer #2 (Recommendations for the authors):1. As summarized in my public comments, my primary concern is lack of clarity in why not all cancer PRS were carried forward from Supplemental Table 1 to evaluation in Supplemental Table 2. The PRS from the largest prostate cancer GWAS to date (Conti 2021), including 12% Asian individuals, is represented in Supplemental Table 1 (PGS ID: PGS000662) but is not carried forward for evaluation in Supplemental Table 2. Similarly, a lung cancer PRS developed specifically for a Chinese population (Dai 2020) is also represented in Supplemental Table 1 (PGS ID: PGS000070) but not carried forward for evaluation. I have not performed similar inspection for the breast and colorectal cancer PRS, but this raises the concern about incomplete evaluation of available (and even best) PRS.

We have included the analysis of the PRSs that did not have training/ testing datasets mentioned and added the results to Supplementary file 1c-f. Thus, we studied a total of 165 PRSs (87 for breast cancer, 26 for colorectal cancer, 13 for lung cancers and 39 for prostate cancer). Figure 1 is updated to the PRSs from this list – PGS000873 (Breast), PGS000662 (Prostate), PGS000055 (Lung-Female), PGS000734 (Lung-Male), PGS000721 (Colorectal-Female), and PGS000070 (Colorectal-Male).

2. It is not clear that AUC should be used to identify the "best" PRS for each cancer in this population, particularly since from what I can tell the author did not use a time-dependent AUC method and excluded all baseline cases from analyses. A time-to-event metric would seen to be a more appropriate way to identify the "best" PRS in these analyses, either using an AUC method for incident event data or using risk association.

To obtain a time-to-event metric for AUC at 5-year, we used AUC.cd() from the survAUC package in R (10.1186/s12874-017-0332-6). The best PRSs chosen were the same except for female colorectal cancer (PGS000149). However, the AUC at 5-year was 0.66677 for PGS000055 (the chosen PRS by AUC from the logistic model), 0.00005 lower than for PGS000149 (surv.AUC = 0.66682). We find the difference non-informative and have presented the AUCs from the Cox proportional hazards model in Supplementary file 1f.

In Discussion (second last paragraph): “We further tested the sensitivity of the PRS selection using a time-to-event metric for AUC (at 5-year), and the differences found were non-informative from the logistic regression (Supplementary file 1g).”

3. Although the term "predictive ability" is used in the landmark Wand 2021 paper (ref 31), I recommend using more specific terms in this manuscript, starting with the abstract (line 66). I think the correct terms would be "relative risk association" and "absolute risk association," as appropriate. Still, Ref 31 (Wand 2021) is helpful and should be introduced in the Introduction to orient to reader to the reporting framework the authors use in presenting this work.

We have indicated the reporting framework used in the last sentence of the Introduction:

“In this study, we evaluated the utility of common PRS, curated in the Polygenic Score (PGS) Catalog, in predicting the risk of the commonly diagnosed cancers with high genetic predisposition (breast, prostate, colorectal, and lung) in a prospective cohort comprising 21,694 participants of East Asian descent in Singapore. The reporting framework recommended for the interpretation and evaluation of PRS detailed in Wand et al. is used (10.1038/s41586-021-03243-6).”

In Results: “PRS predictive ability” replaced with “PRS absolute risk association”

In Results (section “PRS distribution”): “Figure 1 depicts the (A) distribution, (B) discrimination, (C) absolute risk association, and (D) calibration of the best-performing PRS (based on AUC) (Additional file 1 – Supplementary Table 3) for the four cancers studied: breast (PGS000873), prostate (PGS000662), colorectal (female: PGS000055; male: PGS000734), and lung (female: PGS000721; male: PGS000070).”

In Discussion (fourth paragraph): “In accordance with published Polygenic Risk Score Reporting Standards, we reported PRS distribution, discrimination, absolute risk association, and calibration for each of the four common cancers studied [31].”

Caption: “Figure 1. Site-specific polygenic risk scores (PRS) performance assessment.

(A) Distribution, (B) discrimination, (C) absolute risk association and (D) calibration for each of the four common cancers studied. Two-sided, two-sample t-tests with a type I error of 0.05 were used to examine whether there was a difference in the distribution of standardised PRS (subtraction of mean value followed by the division by the standard deviation) between site-specific cancer cases and non-cancer controls (A). The PRS showcased are the best-performing scores based on Area Under the Receiver Operator Characteristic Curve (AUC) values in the female and male populations, (i) unadjusted [solid line], and (ii) adjusted for age at recruitment [dashed line] (B). Each colored line in the plots for absolute risk association denotes a five percentile increase in the standardised PRS score in (C). Calibration calculated based on five-year absolute risk by PRS deciles in (D). A prediction tool is considered more accurate when the AUC is larger. An AUC of 0.9–1.0 is considered excellent, 0.8–0.9 very good, 0.7–0.8 good, 0.6–0.7 sufficient, 0.5–0.6 bad, and less than 0.5 considered not useful (PMID: 27683318).”

4. Can the authors justify why they performed only sex-stratified analyses for lung and colorectal cancer? The GWAS for these conditions included both sexes. Given the relatively small number of cases, it would be worth performing all analyses in the cohort overall. Sex-stratified analyses could be presented in the supplemental materials, if desired.

Gender differences in colorectal and lung cancer incidence have been reported in Singapore.

From Kok et al. (DOI: 10.1007/s00384-007-0421-9): “Male colorectal cancer rates between 1968 and 2002 from 20 to 40 per 100,000 person years. The increase was sharpest among older men, for whom there was a significant AC effect. Female colorectal cancer rates increased until 1992 (from 16 to 29 per 100,000 person years) and stabilized afterward.”

From Lim et al. (DOI: 10.1016/j.lungcan.2014.01.007): “Lung cancer incidence rates were more than two times higher in males compared to females.”

The AUCs associated with selected PRS for combined colorectal (PRS000055, 0.65 [0.63 to 0.67]) and combined lung (PRS000070, 0.69 [0.67 to 0.71]) were not appreciably different from the PRS selected for the sex-stratified analyses (female colorectal: PRS000055, 0.65 [0.62 to 0.69]; male colorectal: PRS000734, 0.66 [0.63 to 0.69]; female lung: PRS000721, 0.69 [0.65 to 0.73]; male lung: PRS000070, 0.68 [0.65 to 0.71]). We have presented the combined-sex analysis for colorectal and lung cancer in Supplementary file 1g.

In Discussion (second last paragraph): “To increase the number of events, we combined the males and females for lung and colorectal cancers. The resulting AUCs (from the logistic regression) were not appreciably different from the sex-specific analysis (Supplementary file 1g).”

5. Lines 182-184: How is it known that only <1% of participants migrated out of Singapore? Are other data available to confirm their ongoing residence in Singapore and absence of cancer diagnosis? E.g., other medical diagnoses or other data?

The data on migration in our cohort was collected during our subsequent follow-up interviews, and we were informed about the migration by the family members of cohort members who had migrated. To our best knowledge, we only knew 47 participants who had migrated to other countries in our cohort for the duration of this study.

In Methods (section “Follow-up”): “The data on migration in our cohort was collected during our subsequent follow-up interviews, and we were informed about the migration by the family members of cohort members who had migrated.”

We have systematically done linkage analysis between our cohort data and the nationwide Singapore Cancer Registry to identify incident cancer cases in our cohort since recruitment in the 1990s. The nationwide cancer registry has been in place since 1968 and has been shown to be comprehensive in its recording of incident cancer cases [reference: Bray F, Colombet M, Mery L, Piñeros M, Znaor A, Zanetti R and Ferlay J, editors (2017). Cancer Incidence in Five Continents, Vol. XI (electronic version) Lyon, IARC. http://ci5.iarc.fr last accessed on 2 December 2022]. Furthermore, compulsory reporting of cases to Singapore Cancer Registry has been mandated by law since 2010. In addition, we were able to ascertain survival status of all cohort participants via record linkage with the population-based Singapore Registry of Births and Deaths. Hence, we are confident that the capture of cancer cases via linkage with national registry can be considered to be virtually complete.

In Methods (section “Selection of common cancers”): “The Singapore Cancer Register was founded in 1968. Prior to 2009, reporting of neoplasms by all medical practitioners and pathology laboratories to the registry is voluntary (10.1016/j.canep.2016.06.006). The registry's staff compares cancer patient hospital discharges and death certificates to registered cases for verification. Completeness of reporting in the 1970s is 96% and in the 1990s, it was close to 100% (10.1186/1471-2407-6-261).”

6. It is reassuring to see cigarette smoking as a strong lung cancer risk factor, but the apparent protective effect of smoking on prostate cancer contradicts current evidence. The authors should discuss this in the Discussion.

We have added in the Discussion:

In Discussion (fifth paragraph): “It is reassuring to see tobacco smoking is a strong risk factor for lung cancer in our dataset. However, smoking appeared to be associated with a protective effect for prostate cancer. While smoking is a well-known risk factor for many cancers (10.18632/oncotarget.24724), in particular lung cancer, observational studies frequently show that smokers are associated with a lower incidence of prostate cancer (10.1038/bjc.2012.520, 10.1158/1055-9965.EPI-09-0252, 10.1002/(SICI)1097-0215(19960917)67:6<764::AID-IJC3>3.0.CO;2-P, 10.1054/bjoc.1999.1105, 10.1007/BF00051881, 10.2105/AJPH.2008.15050, 10.1016/j.eururo.2014.08.059, 10.1186/s12916-016-0607-5, 10.1002/ijc.22788). However, a Mendelian randomisation study did not support the association (10.1371/journal.pmed.1003178).”

7. Lines 274-292 could be shortened, simply referring the reader to most of these results in Table 1.

We have removed the second paragraph under Characteristics of the study population, keeping only details relevant to numbers of incident cancers and age of diagnosis:

In Results (section “Characteristics of the study population”): “Table 1 shows the characteristics of the 21,694 participants who were cancer-free at recruitment. The median follow-up time for the cohort was 20 years (IQR: 18 to 22). As of December 2015, 495 women developed breast cancer, 308 men developed prostate, 774 (332 women and 409 men) colorectal cancer, and 562 (181 women and 381) lung cancer. The median age at recruitment was 54 years (interquartile range [IQR]: 49 to 61). The median age at diagnosis was 65 years (IQR: 59-70) for female breast cancers, 72 years (IQR: 67 to 77) for prostate cancers, 71 years (IQR: 65 to 76) for male colorectal cancers, 71 years (IQR: 64 to 78) for female colorectal cancers, 74 years (IQR: 68 to 78) for male lung cancers and 74 years (IQR: 66 to 79) for female lung cancers.“

8. Lines 368-387: This section of the results seems out of order compared with the Methods. I recommend reorganizing the Results as follows: "Characteristics of the study population," "Lack of Asian representation in PRS development," "PRS discriminatory ability," "PRS distribution," "Associations between PRS and relative hazard of developing cancers," "Number of cancers that developed within PRS at-risk groups," "Association of PRS with absolute risk," "PRS calibration."

We have reordered the sections as suggested without the section on “Number of cancers that developed within PRS at-risk groups” (based on the comments by Reviewer 1).

9. Calibration should be formally tested (e.g., with a Hosmer-Lemeshow or more sophisticated test) in addition to the visual inspection presented.

We did not observe any lack of calibration based on the Hosmer-Lemeshow test (using 10 groups) for all PRSs except PGS000662 (prostate cancer). However, males in the first 5 deciles of PGS000662 did not develop prostate cancer. This may indicate that a linear fit is not appropriate, but a hard threshold (here the start of the 6th decile) is more appropriate to indicate males at elevated risk of developing prostate cancer.

In Methods (section “PRS calibration”): “In addition, we used the Hosmer-Lemeshow test to check the goodness-of-fit.”

In Discussion (paragraph 7): “All PRSs except PGS000662 (prostate cancer) passed the formal Hosmer-Lemeshow goodness-of-fit test. Males in the first 5 deciles of PGS000662 did not develop prostate cancer, suggesting that a linear fit may not be appropriate. A hard threshold beginning from the 6th decile may perform better at identifying males at elevated risk of developing prostate cancer.”

10. Lines 470-476 of the Discussion stray a little from the focused thesis of this work.

We have removed these lines from the Discussion.

[Editors' note: further revisions were suggested prior to acceptance, as described below.]

The manuscript has been improved but there are some remaining issues that need to be addressed, as outlined below:1. Your response to the initial critique #1 is valid. The response itself contains errors (e.g. incorrect PGS numbers given for specific disease PRS), but the supplemental tables appear correct. The caption for Figure 1 is missing, which should explain which column corresponds to which disease and in which sex.

We corrected the errors in the response for the PGS numbers, table numbers, and supplementary files numbers.

“Thus, we studied a total of 165 PRSs (87 for breast cancer, 26 for colorectal cancer, 13 for lung cancers and 39 for prostate cancer). Figure 1 is updated to the PRSs from this list – PGS000873 (Breast), PGS000662 (Prostate), PGS000055 (Lung-Female), PGS000734 (Lung-Male), PGS000721 (Colorectal-Female), and PGS000070 (Colorectal-Male).”

Caption for figure 1:

“Figure 1. Site-specific polygenic risk scores (PRS) performance assessment.

(A) Distribution, (B) discrimination, (C) absolute risk association and D) calibration for each of the four common cancers studied (columns from left to right: breast, prostate, lung [female], lung [male], colorectal [female], and colorectal [male]. Two-sided, two-sample t-tests with a type I error of 0.05 were used to examine whether there was a difference in the distribution of standardised PRS (subtraction of mean value followed by the division by the standard deviation) between site-specific cancer cases and non-cancer controls (A). The PRS showcased are the best-performing scores based on Area Under the Receiver Operator Characteristic Curve (AUC) values in the female and male populations, (i) unadjusted [solid line], and (ii) adjusted for age at recruitment [dashed line] (B). Each colored line in the plots for absolute risk association denotes a five percentile increase in the standardised PRS score in (C). Calibration calculated based on five-year absolute risk by PRS deciles in (D). A prediction tool is considered more accurate when the AUC is larger. An AUC of 0.9–1.0 is considered excellent, 0.8–0.9 very good, 0.7–0.8 good, 0.6–0.7 sufficient, 0.5–0.6 bad, and less than 0.5 considered not useful (PMID: 27683318).”

2. Please add the methods of your new time-to-event AUC sensitivity analysis to the main Methods of the manuscript, including a reference to the R package used. Also please state explicitly in the Methods your approach to choosing the best PRS for each disease (or disease-sex combination): logistic regression AUC.

We added the information in Methods as recommended.

In Methods (PRS discrimination):

“The site-specific PRS with the highest AUC (logistic regression models) was selected. To test the sensitivity of the PRS selection, we obtained a time-to-event metric for AUC at 5-year, we used AUC.cd() from the survAUC package in R (10.1186/s12874-017-0332-6).”

3. Please explicitly state this rationale for sex-stratified analyses for these cancers in the Methods.

We have added the information as recommended.

In Methods (Selection of common cancers)

“We further stratified the analysis by sex as differences in colorectal and lung cancer incidence by sex have been reported in Singapore [10.1007/s00384-007-0421-9].”